# DATA-EFFICIENT GENERALIZATION AND FASTER INITIAL LEARNING IN QUANTUM MODELS FOR CLASSIFYING CELLULAR ACTIVATION STATES

## ABSTRACT

Quantum computing is in its infancy. While it promises to solve some of the intractable problems of computing, real world application is scarce. It is mainly challenged by the hardware which are currently limited both in circuit width and depth. Finding a real world application with an advantage compared to classically available solutions is even harder in the current state-of-the-art machines. However, given the vastly different nature of quantum computers, it is possible the advantage may come from unexpected corners when applied to wide range of classical problems. Machine learning using quantum algorithms are of particular interest due to their ease of parameterization and possible resource efficiency. In this work, we apply a quantum machine learning (QML) algorithm to real world data and benchmark some of the well established scaling laws in a resource constraint scenario using both ideal and noisy ion trap quantum computer platform. The real world problem we investigated comes from the accurate identification of cytotoxic CD8+ T cell activation states from high-dimensional cytometric data. Hand-engineered features extracted from imaging flow cytometry capture morphological, intensity, texture and shape descriptors that are essential for discriminating between quiescent and stimulated cellular states. Leveraging a dataset of processed blood cell images from three patients, we compare quantum data re-uploading classifiers (QDRCs) with classical feedforward neural networks (FNNs) for the task of binary classification of cellular activation. The study is driven by three findings: (1) both quantum and classical models achieve high test accuracy ($\approx 99\%$) when trained with sufficient data and epochs, and models trained on one patient generalize well to the other two, demonstrating the learnability of the engineered feature space; (2) the generalization error of QDRCs exhibits a predictable power-law scaling with training size consistent with a $\sqrt{\frac{T}{N}}$ bound for T trainable parameters, whereas FNNs lack a comparable scaling relationship; and (3) QDRCs converge to high accuracy in early epochs under low-data constraints, aligning with a convex kernel interpretation of the re-uploading model. We further validate a theoretical bound derived from quantum generalization theory and provide an intuitive proof under a convexity assumption. These results indicate that quantum architectures can be competitive with classical baselines while offering faster early generalization and theory-consistent behavior in data-limited regimes, although our conclusions are restricted to hand-crafted features and do not imply clinical readiness or broader generalization.

## 1 INTRODUCTION

Recent developments in quantum computing have spurred interest in quantum machine learning (QML) models, which use parameterized quantum circuits to generate non-classical feature maps or learn unitary transformations. Among these, quantum data re-uploading classifiers (QDRCs) employ a small number of qubits and repeatedly encode input features into quantum states through rotation gates. Classical models such as feedforward neural networks (FNNs), by contrast, learn both the feature representation and classifier through back-propagation on a non-convex loss land-

scape. Theoretical work analysing QML has shown that the generalization error of a QDRC with $T$ trainable gates scales at worst as $\sqrt{\frac{T}{N}}$ when trained on $N$ data points Caro et al. (2022). This scaling suggests that small quantum circuits with few parameters may generalize well in low-data regimes, provided the quantum feature map is sufficiently expressive, whereas classical neural networks lack a comparable bound because their effective dimension can grow with depth and width, making theoretical error control more difficult. Moreover, Quantum Data Re-uploading algorithm has been shown to require much fewer training resources as compared to other quantum models Jerbi et al. (2023).

The potential for QML models to generalize well from few examples makes them ideal candidates for problems where data is high-dimensional yet scarce, a common scenario in biomedical analysis. Imaging flow cytometry is a powerful platform that enables high-throughput acquisition of multi-channel images of individual blood cells. Each cell can be described by dozens of object-level features measuring size, shape, intensity and texture Dominical et al. (2017). For example, the IDEAS® Image Data Exploration and Analysis Software computes numerous base features per image channel, grouped into size, location, shape, texture and signal categories Lippeveld et al. (2020). These measurements are crucial for identifying subtle morphological changes that reflect distinct cellular states, such as cellular activation. For example, during infection or tumor progression, stimulation with inflammatory signals like TNF$\alpha$, can induce immune cells to exhibit altered size, shape, or protein localization patterns compared with their quiescent counterparts. Rapid and accurate AI/ML-enabled classification of such cellular states can provide valuable insights for clinical diagnosis and therapeutic monitoring across a range of conditions, including chronic inflammation, cancer, and acute infections, without the need for time- and cost-intensive staining. However, the high dimensionality of the feature vectors relative to the small number of labeled samples poses a statistical challenge. Standard discriminative models risk overfitting when training data are scarce, motivating the exploration of algorithms that generalize well from few examples.

In this paper, we conduct a comparative study between Quantum Data Re-uploading Classifiers (QDRCs) and classical Feedforward Neural Networks on the task of classifying TNF$\alpha$-stimulated cells from high-dimensional cytometric features. Our work makes three primary contributions. First, we provide a comprehensive empirical benchmark, demonstrating how quantum models perform relative to classical baselines in both data-rich and data-limited scenarios, including under realistic simulated hardware noise. Second, we present an empirical validation of recent theoretical bounds on QML generalization error, testing their predictions on a complex biomedical dataset. Finally, we analyze the learning dynamics of both model families, revealing a notable faster early learning behavior in the quantum models. Furthermore, we provide a theoretical argument to explain this phenomenon.

## 1.1 FEATURE ENGINEERING WITH IDEAS SOFTWARE

Imaging flow cytometry produces rich, multi-channel morphological descriptors that are informative yet statistically challenging when labeled data are scarce. This high-dimensional, low-$N$ setting is a natural testbed. The IDEAS software package produces an extensive set of object-based features from each cytometric image, spanning size (area, perimeter), shape (aspect ratio, compactness), intensity (mean, max), texture (contrast, entropy) and signal location Lippeveld et al. (2020). Importantly, both morphological measurements and intensity information are available for calculating feature values Amn (2009). These features are engineered by domain experts to capture salient aspects of cell morphology and sub-cellular structure. In our study we extract 76 such features per cell, following the same pipeline used in the IDEAS® manual. Because these descriptors incorporate expert knowledge of cellular biology, they provide a strong foundation for classification even with simple models. Given that the feature dimension is high relative to labels, we study models where capacity is explicit in the parameter count $T$; QDRCs realize a fixed feature map with a small trainable readout, which aligns well with our scaling tests.

## 1.2 THEORETICAL MOTIVATION FOR FASTER LEARNING DURING EARLY EPOCHS

To interpret the observed early-epoch learning behavior of quantum models under low-data constraints, we view a QDRC as a fixed feature map $\phi(\cdot)$ followed by a linear readout. Let $\{(x_i, y_i)\}_{i=1}^{N}$,

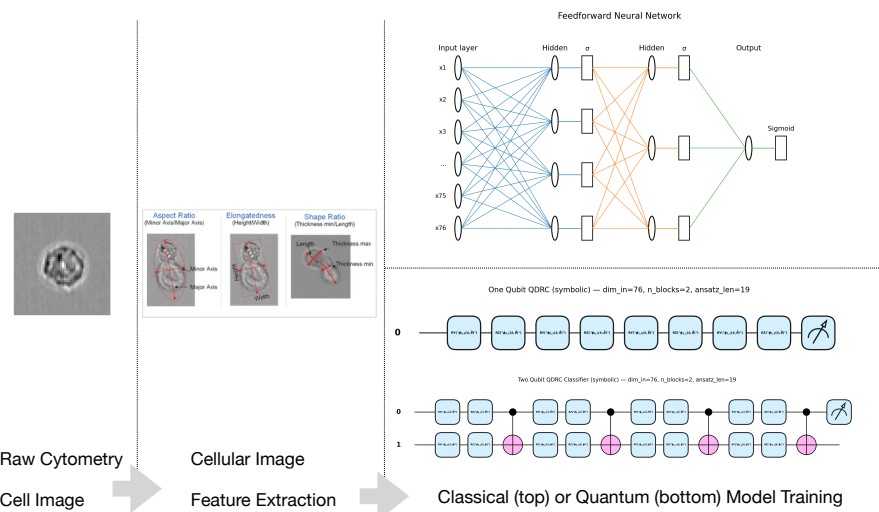

Figure 1: Schematic diagram of the classifier: (left and middle column) The raw cytometry data are processed to extract high dimensional features. The features can then be fed into a feedforward neural network (right top) or a quantum circuit with one or more qubits (right bottom) employing the quantum data re-uploading algorithm for classification of the cells based on their activity.

$y_i \in \{\pm 1\}$, and let $\phi : \mathbb{R}^d \to \mathbb{R}^T$ denote the fixed feature map realized by a QDRC. Write $\Phi \in \mathbb{R}^{N \times T}$ for the design matrix with rows $\phi(x_i)^\top$ and $G := \frac{1}{N} \Phi^\top \Phi$.

**Assumptions.** (A1) $G$ has $\lambda_{\min}(G) = \mu > 0$ on the span of $\{\phi(x_i)\}$ (no degenerate directions at the data). (A2) Step size $\eta \leq 1/\lambda_{\max}(G)$. (A3) The readout is linear with parameters $w \in \mathbb{R}^T$.

**Lemma (linear rate under squared loss).** Bubeck et al. (2015) Under (A1)-(A3), gradient descent on squared loss satisfies

$$\|w_t - w^\star\|_G \leq (1 - \eta\mu)^t \|w_0 - w^\star\|_G.$$

**Local PL (Polyak–Łojasiewicz) for early-epoch regime.** Karimi et al. (2016) Let $z_i(w) = w^\top \phi(x_i)$ and $\sigma(u) = 1/(1 + e^{-u})$. The Hessian of the empirical cross-entropy is

$$\nabla^2 \mathcal{L}(w) = \frac{1}{N} \Phi^\top D(w) \Phi, \qquad D(w) = \text{diag}\big(\sigma(z_i(w))(1 - \sigma(z_i(w)))\big).$$

For any radius $R$ such that $|z_i(w)| \leq c$ for all $i$ whenever $\|w\| \leq R$, we have

$$\sigma(z_i)(1 - \sigma(z_i)) \geq \gamma(c) := \sigma(c)\big(1 - \sigma(c)\big) > 0,$$

hence on the ball $\{\|w\| \leq R\}$,

$$\nabla^2 \mathcal{L}(w) \succeq \gamma(c)\,G, \qquad \text{with } G = \frac{1}{N}\Phi^\top \Phi.$$

If $\lambda_{\min}(G) = \underline{\lambda} > 0$ on the data span, then $\mathcal{L}$ satisfies a Polyak–Łojasiewicz (PL) inequality on that ball with parameter $\mu = \gamma(c)\,\underline{\lambda}$, and gradient descent with step size $\eta \leq 1/L$ enjoys

$$\mathcal{L}(w_t) - \mathcal{L}^\star \leq (1 - \eta\,\mu)^t \big(\mathcal{L}(w_0) - \mathcal{L}^\star\big), \qquad L \leq \frac{1}{4}\,\lambda_{\max}(G).$$

*Early-epoch validity.* With small or zero initialization, $z_i(w_0) = 0$ so $\gamma(0) = \frac{1}{4}$, and for sufficiently small step sizes the iterates remain in a region where $|z_i(w_t)| \leq c$ for the first several epochs, yielding geometric decay with rate parameter $\mu = \gamma(c)\,\underline{\lambda}$. As training progresses and some $|z_i|$ grow, $\gamma(c)$ decreases, explaining the observed slowdown and FNN catch-up at later epochs.

**Remark on capacity.** When $T \geq N$ or features are collinear, $\mu$ can be near zero, slowing the rate and explaining late-epoch catch-up by FNNs at larger $N$.

## 1.3 OUTLINE OF THE PAPER

The remainder of this paper is organized as follows. Section 2 describes the experimental setup, including data collection, feature extraction, model architectures and training procedures. Section 3 presents our results, highlighting three findings: high test accuracy and cross-patient generalization for both quantum and classical models; predictable generalization error scaling in QDRCs but not in FNNs; and an early epoch faster learning behavior of QDRCs under low data. Section 4 discusses the broader implications, limitations and future directions of this comparative study.

## 2 EXPERIMENTAL SETUP

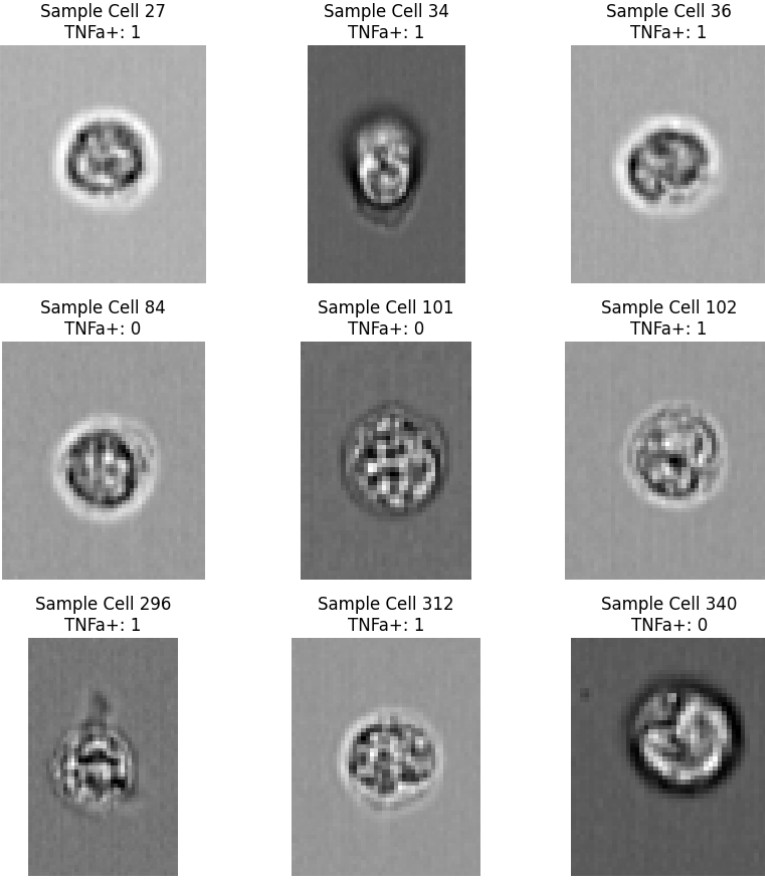

Figure 2: Example cytometry cell images for Patient A. TNF-$\alpha$ (tumor necrosis factor alpha), is a key pro-inflammatory cytokine released during the immune response, typically triggered by infection, tissue damage, or immune activation.

### 2.1 DATA COLLECTION AND FEATURE EXTRACTION

We analyzed imaging flow cytometry data from three individual patients, Fig 2. For each patient, $\approx 1.3$ million of blood cells were processed through a flow cytometer equipped with the IDEAS software. The raw images were segmented to isolate individual cell masks, and 76 quantitative features were computed per cell. These features, as shown in table 1, cover size (e.g., area, perimeter), location (centroid coordinates), shape descriptors (aspect ratio, eccentricity, circularity), intensity measures (mean pixel intensity, standard deviation), and texture descriptors (contrast, entropy, energy). Both morphological and intensity-based measurements are included, providing a rich representation of each cell. The feature names and extraction procedures are detailed in the IDEAS User

Manual, but readers should note that this domain knowledge is proprietary to the imaging platform and not easily replicated by generic convolutional networks.

| Category | Definition | Example features |
|---|---|---|
| Size Descriptors | Features quantifying the physical dimensions of the cell or its components. | Area; Diameter; Length; Perimeter |
| Shape Descriptors | Features that describe the morphology and form of the cell. | Aspect Ratio; Circularity; Compactness; Elongatedness |
| Texture Descriptors | Features that measure the spatial variation of pixel intensities, capturing the granularity and pattern within the cell. | Contrast; Gradient RMS; Modulation; Spot Count |
| Signal Intensity Descriptors | Features quantifying the brightness of fluorescent signals. | Intensity; Max Pixel; Mean Pixel; Median Pixel |

Table 1: Example descriptor categories we extracted from cell image for analysis.

## 2.2 MODEL ARCHITECTURES

### 2.2.1 QUANTUM MODELS

We implemented quantum data re-uploading classifiers using pytorch and pennylane simulation. The simulations are based on both noiseless and noisy circuits. The noisy circuits are implemented with the same noise model as in case of our earlier work on real quantum hardware based on ion trap Yum et al. (2017); Dutta et al.. Three QDRC architectures are considered: a single-qubit classifier without noise, a single-qubit classifier with realistic noise, and a two-qubit classifier without noise. Input features were encoded into rotation gates, with re-uploading layers stacked to increase expressivity. The number of trainable parameters $T$ depended on the number of re-uploadings and qubits. For noisy simulations we added bit-flip (0.5%), phase-flip (0.5%) and depolarizing noise (1%) per layer to mimic the $^{138}\text{Ba}^+$ trapped-ion system used in our laboratory. We will provide an illustration of the single-qubit DQRC, as shown in fig 1, below. The multi-qubit ones are constructed similarly with additional entangling gates such as CNOT.

**Hyperparameters and shapes.** We begin with a single qubit in ground state $|0\rangle$, input dimension $d = \texttt{dim\_in}$, number of blocks $B = \texttt{n\_blocks}$, segment length $L = \texttt{ansatz\_len}$, and hence $T = B(L+1)$ trainable parameters. Define

$$S := \left\lceil \frac{d}{L} \right\rceil, \qquad P := \left\lceil \frac{S}{2} \right\rceil.$$

Parameters per block $k \in \{0, \ldots, B-1\}$:

$$W_k \in \mathbb{R}^d, \qquad b_k \in \mathbb{R}^S.$$

**Input padding and segmentation.** Given $x \in \mathbb{R}^d$, pad to $\tilde{x} \in \mathbb{R}^{SL}$ by zeros:

$$\tilde{x}_j = \begin{cases} x_j, & 0 \le j < d, \\ 0, & d \le j < SL. \end{cases}$$

Partition $\tilde{x}$ into $S$ contiguous length-$L$ segments

$$\mathcal{I}_s := \{sL, \ sL+1, \ \ldots, \ sL+L-1\}, \qquad s = 0, \ldots, S-1.$$

**Per-segment affine map to angles.** For each block $k$ and segment $s$ define

$$\theta_{k,s} = b_{k,s} + \sum_{j \in \mathcal{I}_s} W_{k,j} \, \tilde{x}_j \quad \in \mathbb{R}.$$

Split even and odd segments into rotation angles (the added 0's for the $2r + 1 \geq S$ case is to ensure consistent parameter dimensions when $S$ is odd)

$$\psi_{k,r} := \theta_{k,2r}, \qquad \zeta_{k,r} := \begin{cases} \theta_{k,2r+1}, & 2r+1 < S, \\ 0, & 2r+1 \geq S, \end{cases} \qquad r = 0, \ldots, P-1,$$

so each pair $(\psi_{k,r}, \zeta_{k,r})$ parameterizes a $R_y$ then $R_z$ rotation.

**Per-block unitary and full circuit.** For block $k$ the unitary is the ordered product of $P$ pairs

$$U_k = \prod_{r=0}^{P-1} R_z(\zeta_{k,r}) R_y(\psi_{k,r}) = R_z(\zeta_{k,P-1}) R_y(\psi_{k,P-1}) \cdots R_z(\zeta_{k,0}) R_y(\psi_{k,0}).$$

The full depth-$B$ unitary is

$$U = U_{B-1} U_{B-2} \cdots U_0.$$

**State preparation and optional noise.** Start from $\rho_0 = |0\rangle\langle 0|$. If there is no noise, the output state is the pure state

$$\rho_{\text{out}} = U \rho_0 U^\dagger.$$

If noise channels are enabled, insert a completely positive trace-preserving map $\mathcal{N}_{k,r}$ after each pair in block $k$:

$$\rho_{k,r+1} = \mathcal{N}_{k,r}\big(R_z(\zeta_{k,r}) R_y(\psi_{k,r}) \rho_{k,r} R_y(\psi_{k,r})^\dagger R_z(\zeta_{k,r})^\dagger\big), \quad \rho_{k+1,0} = \rho_{k,P},$$

and set $\rho_{\text{out}} = \rho_{B,0}$.

**Observable and readout.** The observable is $H \in \mathbb{C}^{2\times 2}$ Hermitian with default $H = \sigma_z = \text{diag}(1, -1)$. The raw scalar output is the expectation

$$y = \text{Tr}\big(H \rho_{\text{out}}\big) = \begin{cases} \langle \psi_{\text{out}}|H|\psi_{\text{out}}\rangle, & \text{noiseless}, \\ \text{Tr}\big(H \rho_{\text{out}}\big), & \text{noisy}. \end{cases}$$

Since the readout from quantum measurement is an expectation value, we can simply treat it as the probability of the predicted label, or optionally apply a sigmoid.

**Parameter counts.** For input dimension $d$, segment length $L$, and $S = \lceil d/L \rceil$ segments, each block $k$ has an affine map with weights $W_k \in \mathbb{R}^d$ and biases $b_k \in \mathbb{R}^S$. With $B$ re-uploading blocks on $Q$ qubits using fixed entanglers, the number of trainable parameters is

$$T_{\text{QDRC}} = Q B (d + S),$$

We fix entangling layers (e.g., CNOT) so they do not contribute trainable angles.

**Noise model.** Noisy QDRCs insert, after each $(R_y, R_z)$ pair, independent bit-flip ($p_X$), phase-flip ($p_Z$), and depolarizing ($p_{\text{dep}}$) channels.

### 2.2.2 CLASSICAL MODELS

Using FNNs as our classical baseline, we varied activations between one to five hidden layers. Activations were selected from {ReLU, Sigmoid, Tanh, SeLU} with output sigmoid for binary classification. With input size $d{=}76$ and hidden width $h$, a one-hidden-layer FNN's trainable parameter count is

$$T_{\text{FNN}} = \underbrace{76h}_{\text{input}\rightarrow\text{hidden}} + \underbrace{h}_{b_1} + \underbrace{h}_{\text{hidden}\rightarrow\text{out}} + \underbrace{1}_{b_2} = 78h + 1.$$

We choose $h$ to match $T_{\text{QDRC}}$ as closely as possible. Concretely, we chose $T \in [80, 800]$ by varying number of hidden layers and hidden dimensions.

### 2.3 TRAINING AND EVALUATION PROTOCOL

A rigorous and fair evaluation protocol was established to compare the models.

**Data Splits and Cross-Validation:** For each of the three patients, the corresponding dataset was split into training, validation, and test sets using a stratified sampling approach to maintain the class distribution. To assess the generalization of learned biological features beyond individual-specific artifacts, we also performed a cross-patient evaluation, where models trained on the data from one patient were tested on the data from the other two.

**Generalization error estimation:** We estimated the generalization error $\epsilon_{gen} := loss_{prediction} - loss_{train}$ of each model as the difference between its train loss and test loss on held-out data at different epochs. For each architecture family, we computed $\epsilon_{gen}$ at varying model sizes $T$ and training sizes $N$ and fitted a log–log regression $\log \epsilon_{gen} = a + b \log(T/N)$. According to the theory, QDRCs should satisfy $b \approx 0.5$ in line with the $\sqrt{\frac{T}{N}}$ bound. The regression coefficient $b$ and correlation coefficient $r^2$ were recorded.

**Constrained Training Regimes and Early Epoch Behavior:** To specifically investigate performance in data-limited scenarios, we considered seven constrained training sizes $N \in \{10, 25, 50, 100, 250, 500, 1000\}$. Faster early learning behavior was characterized by plotting test accuracy over epochs for a fixed small training size and small training epochs.

## 3 RESULTS

### 3.1 ASYMPTOTIC PERFORMANCE AND CROSS-PATIENT GENERALIZATION

| QDRC (1-qubit noiseless) | | | | QDRC (1-qubit noisy) | | | |
|---|---|---|---|---|---|---|---|
| **Train** | **Pat. A** | **Pat. B** | **Pat. C** | **Train** | **Pat. A** | **Pat. B** | **Pat. C** |
| Pat. A | 91.3-93.3 | 93.1-96.3 | 97.2-99.7 | Pat. A | 92.6-97.1 | 94.2-98.8 | 93.6-98.9 |
| Pat. B | 90.4-97.3 | 91.8-100 | 96.8-100 | Pat. B | 92.7-98.4 | 94.7-99.2 | 95.4-99.3 |
| Pat. C | 91.8-93.7 | 94.0-96.6 | 97.5-99.8 | Pat. C | 89.3-94.4 | 91.4-97.8 | 96.4-99.7 |

| QDRC (2-qubit noiseless) | | | | FNN (w/ Sigmoid, Tanh, or SeLU) | | | |
|---|---|---|---|---|---|---|---|
| **Train** | **Pat. A** | **Pat. B** | **Pat. C** | **Train** | **Pat. A** | **Pat. B** | **Pat. C** |
| Pat. A | 94.3-98.5 | 96.1-99.7 | 95.4-99.7 | Pat. A | 90.5-99.8 | 92.7-100 | 94.7-99.9 |
| Pat. B | 95.8-98.6 | 97.1-99.8 | 98.4-99.6 | Pat. B | 97.5-99.5 | 99.2-99.9 | 98.7-100 |
| Pat. C | 92.1-94.4 | 95.9-97.8 | 99.1-99.8 | Pat. C | 96.6-98.8 | 99.0-99.9 | 97.7-100 |

Table 2: Test accuracy of QDRC and FNN architectures across patients A, B, and C. First column shows where the model is trained on. The range indicates minimum and maximum test accuracies across different runs (different weight initializaiton and different training data selection) with different model sizes. We have excluded FNN with ReLU activation as it occasionally failed to learn (lower than 70% final test accuracy).

We first asked whether quantum and classical models can accurately distinguish TNF$\alpha$-stimulated cells from their unstimulated counterparts when provided with sufficient training data. Both QDRCs and FNNs achieved high test accuracies, often exceeding 99% after 100 epochs on the full training sets. Cross-patient evaluation further revealed that models trained on one patient generalize well to the other two, indicating that the engineered features capture patient-independent morphological signatures of cellular activation. Table 2 summarizes representative accuracies for models trained on one patient (rows) and tested on another (columns).

### 3.2 GENERALIZATION ERROR SCALING

To test the theoretical prediction that the generalization error of a quantum model scales as $\sqrt{\frac{T}{N}}$, we plotted the empirical generalization error against the ratio between trainable parameters and training

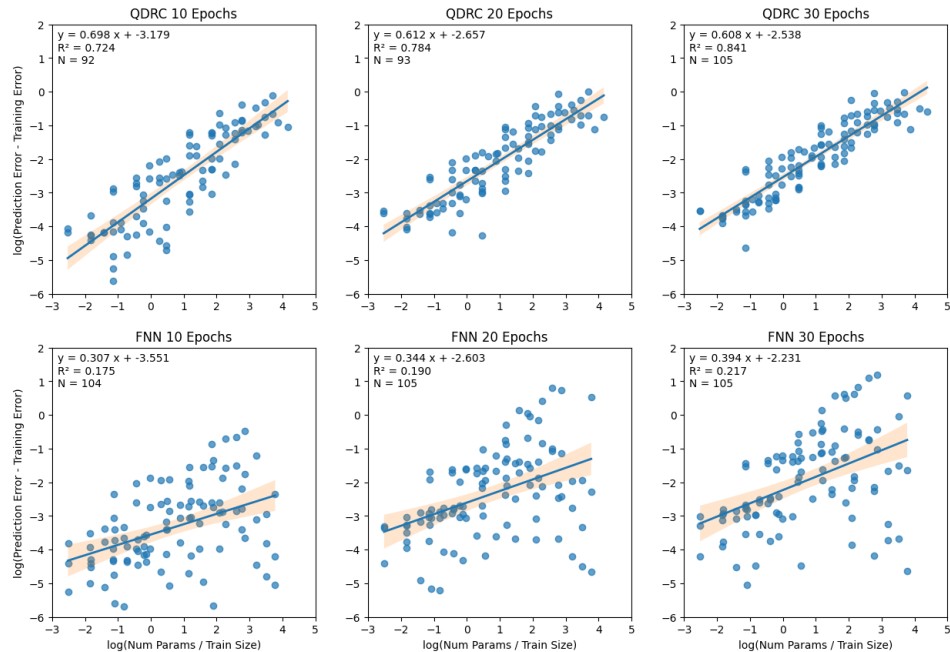

Figure 3: Generalization error scaling of various models trained on Patient B (chosen as a typical data). Log-generalization-error against log-(T/N), where T is the number of trainable parameters and N is the training size. T ranges between 80 to 800 and N ranges between 10 to 1000. The best linear regression is given as $y = ax + b$ and the correlation value is given as $R^2$. Here, N represents the number of points. The shaded band shows the 95% confidence band for the fitted regression line.

size on as a log–log plot in fig 3. The results compare QDRCs with FNNs after 10, 20, or 30 training epochs. For QDRCs the data points lie approximately on a straight line with slope between 0.6 and 0.7, yielding high coefficients of correlation ($R^2 \approx 0.72 - 0.84$). This indicates that the generalization error decreases roughly as $(T/N)^{-s}, s \in (0.6, 0.7)$, close to the theoretical bound, which is asymptotic and may not tightly describe finite-sample behavior. Classical FNNs, however, exhibit no regular pattern: the slope varies between 0.3 and 0.4, and $R^2$ values are markedly lower. These observations support the claim that quantum models obey a predictable scaling law while classical models do not. Note that the absolute generalization error for both model families drops significantly once $Num_{epochs} \geq 100$, explaining the high test accuracies reported above.

## 3.3 Learning in Early Epochs under Data-Limited Regimes

While both model families perform comparably in data-rich settings, significant differences in their learning dynamics emerge under data-constrained conditions. A key finding of this study is the early-epoch performance of QDRCs, particularly when the training set size ($N$) and the model size ($T$) are small. The quantum models reach higher test accuracy faster than their classical counterparts.

This behavior is illustrated in comparative hue plots shown in Figure 4. For small training sets and small models, the average test accuracy of the QDRC models after just 10 epochs is higher than that of the FNN models. The gap gets smaller after 20 epochs and the FNNs eventually catch up and overtake QDRC after further epochs, the QDRCs establish a good performance baseline within few epocs. This performance gap points to a possible learning-efficiency effect for quantum models when data and compute are limited. This is observed across several tested QDRC architectures, including both noiseless and noisy model, suggesting the advantage is a structural property of the classifier's learning dynamics rather than an artifact of idealized conditions.

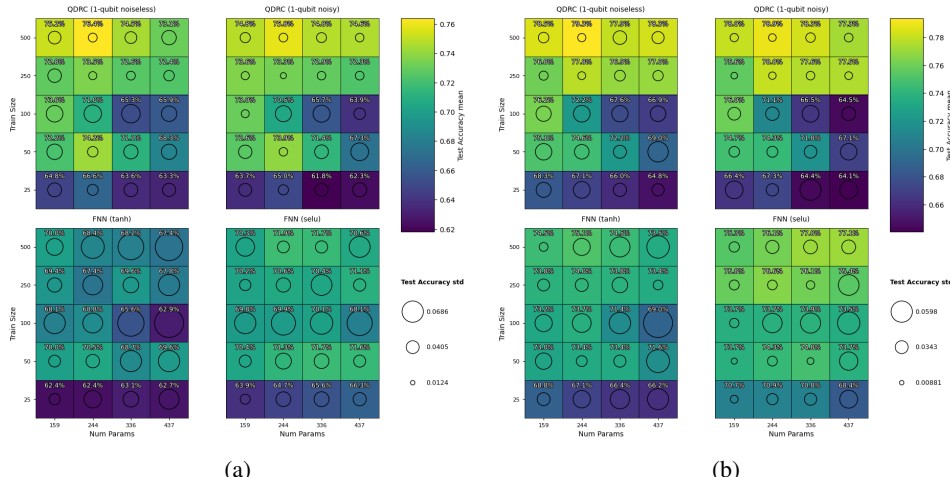

(a)                      (b)

Figure 4: **Limited-resource training:** (a) Constrained training for 10 epochs. X-axis is number of trainable parameters and Y-axis is number of training size. We compared the test accuracy of two quantum and two classical architectures: 1-qubit noiseless QDRC, 1-qubit noisy QDRC, FNN with Tanh activation and FNN with SeLU activation. Each configuration was trained for 50 independent runs with random seeds and we report the numerical test accuracy mean (also shown as hue) and test accuracy standard deviation (as circle radius). (b) Same as (a) but model trained for 20 epochs.

# 4 DISCUSSION

This comparative study explored quantum and classical machine learning approaches to classify cellular activation states based on hand-engineered features. The three main findings are: (1) both QDRCs and FNNs achieve near-perfect accuracy when trained on sufficient data and generalize well across patients, confirming that the 76-dimensional feature space effectively captures the morphological response to TNF$\alpha$ stimulation; (2) the generalization error of QDRCs obeys a predictable power-law scaling with training size, with scaling values close to the theoretical $\sqrt{\frac{T}{N}}$ bound, whereas FNNs lack such a relationship; and (3) quantum models learn faster in early epochs under low-data and small model settings. We attribute the early-epoch learning behavior to a fixed-feature, linear-readout view: in early epochs the logits remain bounded, so cross-entropy satisfies a local PL condition with a rate constant proportional to $\lambda_{\min}(G)$. As logits saturate, this constant shrinks, explaining the later slowdown of quantum models. These results suggest that small quantum circuits can be competitive with classical neural networks while providing theoretical guarantees and rapid learning.

Nevertheless, several limitations warrant caution. First, our dataset comprises only three patients and 76 handcrafted features; extending the analysis to larger cohorts and raw image inputs would be necessary to establish clinical utility. Second, the performance differences observed here are modest and manifest mainly under low-data constraints; with abundant data both model families perform similarly. Third, the noise simulations assume idealized error rates; real quantum hardware may exhibit correlated errors not captured in our model. Finally, our results should not be interpreted as evidence of general quantum advantage over classical computers. Rather, they highlight a specific context—feature-based classification of cellular activation states—where quantum models exhibit favorable scaling and learning dynamics. Future work could investigate hybrid quantum–classical architectures, extend the theory to deep circuits, and evaluate the impact of more complex noise models. With continued advances in quantum hardware and algorithm design, QML may offer practical benefits for biomedical applications, but rigorous benchmarking against classical baselines remains essential.

ACKNOWLEDGMENTS

This research is supported by the National Research Foundation, Singapore under the National Quantum Processor Initiative.

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

## A APPENDIX

**LLM usages.** ChatGPT 5 and Gemini 2.5 Pro are used to help polish the writing, format tables and formula in a more presentable manner, improve logical jumps between sections and paragraphs, as well as finding sources and related works.

