# OpenReview forum: "Data-Efficient Generalization and Faster Initial Learning in Quantum Models for Classifying Cellular Activation States"
_ICLR.cc/2026/Conference — Submitted to ICLR 2026_

### Official Review · Reviewer_RTyp · 2025-11-02

**Soundness:** 3
**Presentation:** 3
**Contribution:** 2
**Rating:** 2
**Confidence:** 3

**Summary:**

This paper investigates the generalization behaviour and learning dynamics of Quantum Data Re-uploading Classifiers (QDRCs) versus classical Feedforward Neural Networks (FNNs) on a biomedical dataset derived from imaging flow cytometry. Each cell is represented by 76 engineered features describing morphology and intensity. The authors compare single- and two-qubit QDRCs (both noiseless and noisy) with classical FNNs matched by parameter count.
They report three main findings:
* Both quantum and classical models achieve high accuracy (~99%) and good cross-patient transfer.
* QDRCs follow a predictable power-law generalization scaling.
* QDRCs show faster early-epoch convergence in low-data regimes.

**Strengths:**

* The work validates a recent quantum generalization bound (Caro et al., 2022) using a real-world dataset, not just synthetic tasks.
* The cytometry task, although small-scale, adds diversity to QML benchmarking and demonstrates potential for quantum models in structured feature data.
* The paper systematically varies dataset size, model capacity, and training epochs, and includes a realistic trapped-ion noise model.
* Insightful analysis of early learning. The theoretical explanation linking early-epoch convergence to convex kernel dynamics is well-articulated and plausible.

**Weaknesses:**

* The study does not introduce new algorithms or architectures; it mainly tests existing QDRCs under known theoretical expectations. The novelty lies in the application and analysis rather than method development.
* Since both models operate on hand-engineered tabular features, this setting is far removed from high-dimensional representation learning problems that interest the ICLR audience.
 * Comparing only against small FNNs is insufficient. Classical baselines such as logistic regression, SVMs, or kernel ridge regression would provide more informative references for data efficiency.
* The early-epoch advantage and slight scaling consistency may not translate into meaningful or lasting performance improvements.
* Results are based on three patients and simulations of at most two qubits. The generalisation claims are therefore limited to toy-scale quantum circuits and may not survive in real devices or larger systems.
* Overstated framing. The title and abstract suggest a broad finding on “data-efficient generalization in quantum models,” yet the results mostly confirm previously known theoretical scaling on a small dataset.

**Questions:**

* Could the early-epoch behaviour reflect an implicit regularisation effect due to limited parameter entanglement rather than quantum dynamics per se?
* Could the faster early-epoch convergence be reproduced by linearized or kernelised classical models (e.g., logistic regression), implying that the effect is not inherently quantum?
* Would the generalization law still hold if the model were trained end-to-end on raw image inputs, not pre-extracted features?

---

### Official Review · Reviewer_ckhB · 2025-11-06

**Soundness:** 2
**Presentation:** 3
**Contribution:** 3
**Rating:** 2
**Confidence:** 3

**Summary:**

This paper presents a comparative study between a Quantum Data Re-uploading Classifier (QDRC) and a classical Feedforward Neural Network (FNN) on a real-world biomedical task: classifying T-cell activation states. The data is described as "high-dimensional" (76 features) and "small-sample" (from 3 patients). The paper's main findings are: (1) both models achieve high accuracy (≈99%) with sufficient data; (2) the QDRC's generalization error follows a predictable $\sqrt{T/N}$ power-law scaling, while the FNN's does not; and (3) the QDRC shows faster initial learning in the first 10-20 epochs under data constraints.

**Strengths:**

The paper evaluates quantum machine learning (QML) for a real-world biomedical classification: differentiating activated vs quiescent CD8⁺ T-cells from high-dimensional imaging flow cytometry features. The authors compare quantum data re-uploading classifiers (QDRCs) with classical feed-forward neural networks (FNNs). Their key findings are: (1) both models achieve ~99 % accuracy with sufficient data and generalize across patients; (2) the generalization error of QDRCs follows a predictable power-law scaling with training size (consistent with a qT/N bound for 𝑇 trainable parameters), whereas FNNs lack such scaling; (3) under low-data constraints, QDRCs converge faster in initial epochs, which the authors attribute to a kernel-like interpretation of the re-uploading model. They validate a theoretical bound (under a convexity assumption) derived from quantum generalization theory. The work suggests quantum architectures can be competitive in data-limited regimes and exhibit favorable learning dynamics, though this is limited to engineered features and small cohorts.

**Weaknesses:**

Major Concerns:

On the Framing of the "Real-World Problem":
A potential concern is the paper's framing of the task as a complex "real-world" classification problem. The models are not processing raw cell images, but rather a 76-dimensional "hand-crafted feature" vector extracted by proprietary software.

This implies that the most challenging feature engineering step has already been completed by domain experts and the software.

This 76-dimensional feature vector appears to be a highly concentrated and well-designed dataset. This might simplify the task from a complex machine learning problem to one of shallow classification on a well-defined feature space.

Questions Regarding the Choice of Classical Baseline (FNN):
The paper's core argument for the QDRC's data efficiency rests on a comparison with an FNN. This choice of baseline raises some questions, as an FNN may not be the most suitable or strongest model for this specific type of "high-dimensional feature, small-sample" task.

In classical machine learning, methods like SVMs, Random Forests, or Gradient Boosted Trees are often the standard and highly effective tools for this kind of "shallow" tabular data.

These classical models (particularly SVMs) are known to be inherently data-efficient in high-dimensional spaces and might achieve very high accuracy with only a small number of samples.

This leads to a key question: how would the QDRC's data efficiency and "faster early learning" compare against a more traditional and potentially stronger baseline like an SVM? Without this comparison, it is difficult to be certain that the observed advantages are unique to the quantum model rather than just a reflection of the FNN's limitations on this task.

Interpretation of "Faster Learning" and "Scaling Laws":

The paper notes that the QDRC's early advantage is temporary, with the FNN eventually "catching up and overtaking" (Sec 3.3). The practical significance of this temporary advantage is unclear, especially if a more standard classical model (like an SVM) could potentially converge to a high accuracy almost immediately.

The verification that QDRC's generalization error fits the $\sqrt{T/N}$ theory is a valuable scientific validation. However, this seems to be evidence of the model's predictability, not necessarily its superiority. The fact that the FNN does not fit this simple law just implies its generalization behavior is more complex, not that its generalization error is inherently worse.

**Questions:**

Some aspects may be improved and clarified,

1. Limited dataset and domain scope

The dataset comprises only three patients and a 76-dimensional hand-engineered feature set rather than raw image or flow-cytometry data. The limited sample size and narrow feature domain raise concerns about generality of the claims (clinical readiness and broader generalization are explicitly excluded by the authors themselves).

Because the features are engineered rather than raw high-dimensional data, the quantum advantage might stem more from the smaller input space than from intrinsic quantum model strengths; thus the relevance to large-scale deep-learning problems (e.g., raw images, millions of features) is unclear.

2. Quantum vs classical comparison caveats

While QDRCs apparently show faster early-epoch convergence and scaling behaviour, the performance advantage is modest, and under abundant data both QDRCs and FNNs perform similarly. This suggests the quantum benefit might only hold in niche low-data regimes.

It is unclear whether the classical baseline architectures were optimised equivalently (in hyperparameters, regularisation, architecture depth) or whether the feature-engineering favoured one class of models. Without strong classical baselines, the claim of quantum competitiveness remains tentative.

3. Theoretical bound assumptions and hardware realism

The theoretical bound for QDRCs (power-law scaling of generalization error) is derived under a convexity assumption, which may not hold for more complex architectures or non-convex training. The authors should make clear the conditions under which the bound applies, and whether those reflect the actual training regime of the QDRCs.

The experiments include “ideal and noisy ion trap quantum computer platform” simulations, but real hardware may include error correlations, drift, connectivity constraints, and other non-idealities which may alter generalisation/learning dynamics significantly. The paper acknowledges this, but the practical readiness for NISQ or fault-tolerant devices is not established.

4. Scalability and feature-space mismatch

While the QDRC exhibits favourable scaling in the given feature-space, how this would extend to high-dimensional raw data, larger numbers of classes, more complex tasks (multi-class, multi-label) remains unaddressed.

The use of hand-engineered features reduces the dimensionality and complexity of the learning task, which could bias the result in favour of simpler models (quantum or classical) and may not reflect real deep-feature learning scenarios.

5. Interpretability of “faster early learning” attribution

The claim that QDRCs learn faster in the early epochs is attributed to a “fixed-feature, linear-readout view” and a PL-condition rate constant proportional to the minimum eigenvalue of G (the Gram/kernel matrix). While plausible, the empirical validation of this mechanism is limited. More diagnostics (such as measured eigenvalues, kernel spectra, trajectory of logits) would strengthen this intuition.

**Relevant References for Inclusion**

To strengthen the manuscript and situate it properly within the QML and generalisation literature, the authors should include the following:

Banchi, L., Pereira, J., & Pirandola, S. (2021). Generalization in Quantum Machine Learning: A Quantum Information Standpoint. PRX Quantum 2, 040321.

Caro, M. C., et al. (2022). Generalization in quantum machine learning from few training data. PMC, (2022).

Khanal, B., Rivas, P., Sanjel, A., Sooksatra, K., Quevedo, E., Rodriguez, A. (2024). Generalization Error Bound for Quantum Machine Learning in NISQ Era – A Survey. arXiv:2409.07626.

Barthe, A. & Pérez-Salinas, A. (2024). Gradients and frequency profiles of quantum re-uploading models. Quantum 8:1523.

---

### Official Review · Reviewer_kCZm · 2025-11-09

**Soundness:** 1
**Presentation:** 2
**Contribution:** 2
**Rating:** 2
**Confidence:** 4

**Summary:**

The paper investigates the use of a quantum-machine-learning classifier, specifically the Quantum Data Re‑uploading Classifier (QDRC), and compares it to classical feed-forward neural networks (FNNs) on a biomedical classification task: identifying cytotoxic CD8+ T-cell activation states from high-dimensional flow-cytometry features (76 hand-engineered descriptors).

The authors report three main findings:

(1) both quantum and classical models can achieve near-perfect accuracy when given sufficient data.

(2) QDRCs exhibit a predictable power-law scaling of generalization error with T/N (number of trainable parameters T and number of training examples N), whereas FNNs do not show a comparable scaling law.

(3) QDRCs converge more rapidly in early epochs under low-data / small-model conditions, which the authors attribute (via a PL-condition analysis) to their fixed-feature-map + linear-readout structure. The paper further discusses the implications, limitations (small cohort of three patients, hand-crafted features, simulated noise models), and suggests future directions (hybrid QC/CC models, deeper circuits, more realistic noise).

**Strengths:**

- The topic is timely and relevant: comparing quantum and classical models on a **real-world** biomedical dataset goes beyond many purely theoretical QML studies.
- Applying a quantum data re-uploading classifier to a biomedical feature-classification problem is novel, and connecting generalization-error scaling and early-epoch learning behaviour with theoretical analysis (PL condition) represents a thoughtful attempt to bridge QML theory and practice.
- The domain application—classifying cellular activation states from high-dimensional cytometry features—is scientifically meaningful, and the authors clearly acknowledge the limitations of current quantum hardware.
- The writing and structure are clear overall, and the discussion section responsibly notes several limitations without overstating "quantum advantage".

**Weaknesses:**

- **Narrow experimental scope.**
  The study is confined to a single biomedical task with three patients and 76 hand-engineered features. While suitable as a proof of concept, this setup is too limited to justify broad claims about scaling or generalization behaviour in quantum models.

- **Underdeveloped classical baseline.**
  The FNN comparison is shallowly explored. Details about optimizer choice, learning-rate schedules, batch sizes, and hyper-parameter tuning are missing. The FNN architecture is varied only coarsely (1–5 hidden layers) with the hidden width h adjusted solely to match the total parameter count T of the quantum models (T ∈ [80, 800]). This coupling of depth and width prevents disentangling how width affects generalization—an aspect known to strongly influence learning dynamics in classical networks.

- **Inconclusive scaling analysis.**
  The claimed T/N power-law scaling for QDRCs relies on a limited range of model sizes and a simple log–log fit (R² ≈ 0.72–0.84). The classical models show a much weaker correlation (R² ≈ 0.17–0.22), which may stem from insufficient exploration rather than a fundamental difference. Without broader architectural sweeps (e.g., explicit width variation at fixed depth), ablations on optimizer and initialization, or tests on other datasets, the scaling claim remains tentative.

- **Early-epoch behaviour likely confounded.**
  The reported "early-epoch advantage" of QDRCs could result from architectural or training-setup differences rather than an intrinsic quantum effect. Wider networks typically converge faster due to smoother gradients (as predicted by NTK theory), while deeper or narrower ones train more slowly. Combined with possible optimizer or hyper-parameter differences, the observed effect cannot yet be attributed confidently to model type.

- **Overstated framing of contributions.**
  In the **Introduction** (last paragraph), the paper describes its study as a "comprehensive empirical benchmark". Given that the experiments involve only one small dataset, this phrasing overstates the scope.

**Questions:**

1. **Classical baseline configuration.**
   Please provide complete details of the FNN setup: optimizer type (e.g., SGD, Adam), learning-rate schedule, batch size, and hyper-parameter tuning procedure. Clarify whether identical or comparable optimizers were used for quantum and classical models.

2. **Statistical reporting.**
   Report mean ± standard deviation of test accuracy and generalization error (not only min–max ranges), and specify the number of independent runs, random seeds, and training/test splits used to assess statistical robustness.

3. **Cross-patient results.**
   In Table 2, models trained on one patient sometimes perform better on other patients than on the same one. Please discuss possible reasons and provide per-patient sample counts and class ratios.

4. **Architectural ablations.**
   Conduct controlled experiments where FNN depth and width are varied independently (rather than co-varied to match T). This would clarify whether scaling and early-epoch differences arise from architectural factors or genuine model-type effects.

5. **Clarification of "benchmark" scope.**
   The **Introduction** (final paragraph) characterizes the work as "a comprehensive empirical benchmark". Consider rephrasing or clarifying this to reflect that the study represents an initial case study rather than a broad benchmark.

**Details Of Ethics Concerns:**

The manuscript includes self-identifying language (“**our earlier work** on real quantum hardware based on ion trap [Yum
 et al.(2017)]; [Dutta et al.]”) that may compromise anonymity under the ICLR double-blind review policy. This should be rewritten in an anonymized form (e.g., “following prior ion trap studies”) to maintain compliance with research integrity and double-blind review standards.

---

### Meta-Review · Area_Chair_c1X1 · 2026-01-07

**Summary:**

The authors did not submit the rebuttal. All three reviewers were unconvinced on the positive side; they agreed that this work requires additional effort to meet the acceptance bar of ICLR. Thus, I am inclined not to accept this draft at this stage. Thank you for your effort. It is an interesting work. I hope the input from the reviewers will help you further improve this work.

**Reviewer Concerns:**

This work has limitations in novelty and experimental results.

**Reviewer Scores:**

The reviewers' scores consistently reflect the limitations of this work.

---

### Decision · Program_Chairs · 2026-01-26

Reject